# Prevalence of Hemorrhagic Complications in Hospitalized Patients with Pulmonary Embolism

**DOI:** 10.3390/jpm12071133

**Published:** 2022-07-13

**Authors:** Nikolaos Pagkratis, Miltiadis Matsagas, Foteini Malli, Konstantinos I. Gourgoulianis, Ourania S. Kotsiou

**Affiliations:** 1Psychiatric Clinic, General Hospital of Corfu, 49100 Corfu, Greece; nikos.pag@hotmail.com; 2Department of Respiratory Medicine, Faculty of Medicine, University of Thessaly, 41110 Larissa, Greece; mimats@med.uth.gr; 3Vascular Surgery Department, Faculty of Medicine, University of Thessaly, 41110 Larissa, Greece; mallifoteini@yahoo.gr; 4Faculty of Nursing, University of Thessaly, 41110 Larissa, Greece; kgourg@med.uth.gr

**Keywords:** pulmonary embolism, venous thromboembolism, bleeding complications, anticoagulant treatment, prediction of bleeding, in-hospital bleeding

## Abstract

Background: The prevalence of anticoagulant therapy-associated hemorrhagic complications in hospitalized patients with pulmonary embolism (PE) has been scarcely investigated. Aim: To evaluate the prevalence of hemorrhages in hospitalized PE patients. Methods: The Information System “ASKLIPIOS™ HOSPITAL” implemented in the Respiratory Medicine Department, University of Thessaly, was used to collect demographic, clinical and outcome data from January 2013 to April 2021. Results: 326 patients were included. Males outnumbered females. The population’s mean age was 68.7 ± 17.0 years. The majority received low molecular weight heparin (LMWH). Only 5% received direct oral anticoagulants. 15% of the population were complicated with hemorrhage, of whom 18.4% experienced a major event. Major hemorrhages were fewer than minor (29.8% vs. 70.2%, *p* = 0.001). Nadroparin related to 83.3% of the major events. Hematuria was the most common hemorrhagic event. 22% of patients with major events received a transfusion, and 11% were admitted to intensive care unit (ICU). The events lasted for 3 ± 2 days. No death was recorded. Conclusions: 1/5 of the patients hospitalized for PE complicated with hemorrhage without a fatal outcome. The hemorrhages were mainly minor and lasted for 3 ± 2 days. Among LMWHs, nadroparin was related to a higher percentage of hemorrhages.

## 1. Introduction

Pulmonary embolism (PE) is defined as a blockage in the pulmonary artery and its branches. It is caused by detached blood clots that move through the large veins to the pulmonary arteries. Embolism is usually caused by blood clots in the deep network of veins of the lower limbs—mainly in their proximal parts—such as by blood clots in the pelvic network, the upper limbs, and the right part of the heart. Rarely PE is caused by nonthrombotic sources, such as amniotic fluid, tumors, fat, large amounts of air and foreign bodies. In every patient suffering from PE, there is a degree of pulmonary obstruction. The effects of the mechanical obstruction depend on the percentage of the pulmonary circulation that is obstructed, the existence or non-existence of a cardio-respiratory disease and on time taken for the obstruction to occur [1]. If the amount of obstruction is higher than 30%, then the pressure in the pulmonary artery is increased well beyond normal, and consequently, the right part of the heart is beaten. A serious obstruction cannot be compensated by pulmonary capillaries, thus leading to increased pulmonary vascular resistance. This, in turn, provokes an increase in the right ventricular afterload, which results in increased parietal tension and finally, ischemia. Respiratory effects include tachypnea in 92% of patients and serious hypoxemia (PaO_2_ < 70%) in a 63% [2].

PE presents a wide range of hemodynamic effects, from asymptomatic and undiagnosed disease to life-threatening emergencies. It is the third most frequent cause of death in hospitalized patients and a major cause of morbidity and mortality, with a total annual effect of 62 to 112 cases per 100,000 inhabitants [3]. Prognosis may worsen in PE patients, during intrahospital treatment, by experiencing hemorrhagic complications, which are mainly attributed to anticoagulant therapy [4].

Even with the best-coordinated care, hemorrhagic complications may occur. A minor hemorrhage could predict a major one and lead to modification of the anticoagulant therapy, underlying its importance for the prognosis and the efficient management of the major hemorrhagic episodes [1]. Hemorrhage is the most frequent complication caused by any anticoagulant [5].

Only a few studies investigated the in-hospital hemorrhage cases in patients with PE. Data regarding the in-hospital hemorrhagic complications in patients with PE presented with hemodynamic instability have also been scarcely noted, while the percentage of hemorrhagic complications has not been clarified in those receiving thrombolytic therapy. It is also important that there are no references regarding minor hemorrhages in patients hospitalized for PE.

In that context, this study aimed to evaluate the prevalence of hemorrhagic events in hospitalized PE patients and investigate the correlation of hemorrhagic events with the type of anticoagulant treatment, patients’ demographic and clinical parameters, clinical burden, and outcome.

## 2. Materials and Methods

### 2.1. Study Participants

This was a retrospective study recording the hemorrhagic complications of patients with confirmed PE who were hospitalized in the Department of Respiratory Medicine of the University of Thessaly from January 2013 to April 2021. This research included all hospitalized patients in the Department of Respiratory Medicine, University of Thessaly with a discharge diagnosis I-26 Pulmonary Embolism (coding in ICD-10).

### 2.2. Data Collection

Demographic, clinical data, the type of anticoagulant treatment, the burden of disease, hemorrhagic events and outcomes were recorded by the Health Information System “ASKLIPIOS™ HOSPITAL” of the University Hospital of Larissa. Overview of all parameters extracted from the recordings are presented in Table 1.

### 2.3. Statistical Analysis

The chi-square test was used to make comparisons between frequencies. Unpaired t-tes was used for comparing parametric data between two groups, while non-parametric data were analyzed with the Mann–Whitney U test. Parametric data comparing three or more groups were analyzed with one-way ANOVA and Tukey’s multiple comparisons test, while non-parametric were analyzed with the Kruskal–Wallis test and Dunn’s multiple comparison test. Spearman’s correlation was used for correlation analysis. Multiple logistic regression was used to examine a series of predictor variables to determine those that best predict a hemorrhagic event. Statistical analyses were performed with IBM SPSS Statistics for Windows, version 23.0, IBM Corp., Armonk, NY, USA.

## 3. Results

The study included 326 patients with a mean age of 68.7 ± 17.0 years. 57.7% (188) of them were men and much younger than the women. 97.5% of patients were Greek, 1.2% were refugees, and the rest 1.3% were of other nationalities. 86.2% of the population had at least one comorbidity, with arterial hypertension being the most frequent one (52.5% of the patients).

Demographics and comorbidities are presented in Table 2. 8.9% of the total population had no prior medical history. 61.1% of men had had recent surgery in the last three months. Three of these operations had been performed on the vertebral column. In females, three cases of PE were noted in the postnatal period, three cases noted after a recent fracture and immobilization, and two cases after a recent fracture.

17.1% had a history of malignancy and 7.3% of them had a gender-related active disease. An absolute predominance of men (3.7% of the total population) was observed in the most frequent malignancy which is lung cancer. Surprisingly, in the present study, cancer was firstly diagnosed in 2.8% of patients, and more specifically, PE was the first sign of malignancy. 25.2% of the patients (82 people) had a history of thrombosis. 17.2% of the population received antiplatelet agents without any difference in the gender noticed. 5.8% (19) of the patients mentioned a previous episode of hemorrhage, and 21.1% presented a new hemorrhage during the hospitalization because of the PE. 9.2% of the population had a history of thrombophilia.

During the hospital admission, 44.2% presented dyspnea, 32% presented thoracic pain, 26.7% presented fever, and 7.7% had bloody sputum, while 4% of the population was asymptomatic. 18% presented tachycardia in the electrocardiogram (ECG).

Wells scores and Geneva scores, as well as the rates of the laboratory on patients’ admission are presented in Table 3. 26% of the population had respiratory failure and 54% had hypocapnia on admission.

3.3% of the population presented with thrombocytopenia on admission. 10% were complicated by a fall in the number of platelets and thrombocytopenia during the hospitalization. 3.3% had an abnormal international normalized ratio (INR) >1.50, and 19.9% presented uremia on admission. 50% of the patients had a proximal deep vein thrombosis (DVT). 1.8% of the population had a paradox embolism.

Most of the patients admitted (92.4%) received LMWH, as shown in Table 3. 74.7% of them received 12-h action LMWH, and the rest received one subcutaneous dosage daily.

The anticoagulant therapy administered during patients’ hospitalization is presented in Table 4. In 57.3% of the population, the treatment was modified during hospitalization. Specifically, 51.8% shifted to direct oral anticoagulants (DOACs), with which they were discharged. In 9.2% of the population, there had been a shift from 12-h to 24-h action Low-Molecular-Weight Heparin (LMWH), while in 0.9%, there had been a shift from 24-h to 12-h action LMWH. There was only one case of switching from DOAC to LMWH after an episode of gastric bleeding. The patients more frequently received rivaroxaban (75%) and less frequently dabigatran (12.5%) and apixaban (12.5%).

15% of the hospitalized patients (49 people) experienced an episode of hemorrhage without any gender difference (12.2% of men vs. 17.4% of women, *p* = 0.240). 18.4% of them experienced a major hemorrhage, without any difference regarding the gender noticed. Major hemorrhages were much fewer than the minor ones (18.4% vs. 81.6%, *p* = 0.001), while the average duration of hemorrhage was 3 ± 2 days. The sites of the hemorrhage are presented in Table 5.

16% of the patients with hemorrhagic complications (2.1% of the sum) needed a transfusion. The patients who had been transfused were the ones that presented major hemorrhages. 2.1% of the patients with hemorrhagic complications (0.3% of the total population) needed to be transferred to an ICU because of the bleeding.

4.2% of patients with hemorrhagic complications had to interrupt the anticoagulant therapy by missing doses, and 19.1% had to shift to 12-h action LMWH, especially enoxaparin. One out of the 49 patients with hemorrhagic complications who interrupted the therapy experienced a thrombotic event (2.1%). The average duration of hospitalization was 8 ± 5 days. 5.2% of the patients died. No death due to hemorrhagic complications was recorded.

The highest Wells score and the highest rate of creatinine (1.3 vs. 12 + 0.2, *p* = 0.029) were positively correlated with the risk of hospital bleeding. An accounting regression model was used to search for dependent variables (age, gender, comorbidities, the presence of cancer, Wells score, INR on admission, uremia on admission, location of PE, PESI score, ICU, platelet count on admission, right heart failure, instability, antithrombotics) to identify the parameters that could predict hospital bleeding, but no clinical or laboratory predictors were identified.

## 4. Discussion

The frequency of hemorrhagic complications during the hospitalization of patients with PE has not been previously determined in Greece. The present study was the first to investigate this issue. We found that 15% of the population hospitalized due to PE were complicated with hemorrhage, of whom 18.4% experienced a major event. Major hemorrhages were fewer than minor. Nadroparin related to 83.3% of the major events. Hematuria was the most common hemorrhagic event. 22% of patients with major events received a transfusion, and 11% were admitted to ICU. The events lasted for 3 ± 2 days. No death was recorded.

We found that among the hospitalized patients due to PE, males outnumbered females, a finding following the literature supporting that the risk of PE is higher in men than in women [6]. In some studies, the frequency of unprovoked PE varies between 16.5% and 51%, up to 69–76% [7,8,9]. We considered that unprovoked PE should be accepted when there are no comorbidities or provocations with proven PE hazards. Based on this definition, it was found that the frequency of unprovoked PE was 8.9% in our study. However, major predisposing factors were detected in the majority population, such as major surgery, fractures, and postnatal period [10].

The delayed diagnosis of PE was a finding of great interest that accords with previous reports commenting that PE has no typical symptomatology, thus, confirming the difficulty in PE diagnosis [11]. Patient delay of an average of 4.2 days and delay in primary care of an average of 3.9 days were the major contributors to this delay [12]. However, diagnostic delay of PE of more than seven days is common in primary care, especially in the elderly, and if chest symptoms, like pain on inspiration, are absent [13,14].

Surprisingly, in the present study, cancer was firstly diagnosed in 2.8% of patients, and more specifically, PE was the first sign of malignancy. In the case of cancer, the venous thromboembolism (VTE) risk is increased from 7 up to 28 times [15]. Neoplasia is caused when the tumor secretes substances with a prothrombotic effect, such as adhesion molecules that activate the macrophages and the platelets [16]. It has been reported that cancer is usually diagnosed within the first months after a VTE episode, with an overall incident rate of 4.1% in 1 month and 6.3% in 1 year [17].

25.2% of the patients had a previous history of thrombosis. The location and manifestation of thrombosis are of great predictive value for the risk of re-thrombosis. In a meta-analysis of patients with PE or/and DVT, the re-thrombosis percentages were 22% for PE and 26.4% for DVT [18]. The risk for a new PE was 3.1 times higher in patients with symptomatic PE than in those with proximal DVT. The patients with proximal DVT had a 4.8 times higher percentage of relapse than those with peripheral DVT [19].

In the present study, 3.1% of the patients were suffering from chronic renal failure (CRF), a disease associated with an increased risk for hemorrhage, because of the platelet dysfunction and uremic toxins in the blood, which harms the primary hemostasis [20]. Also, patients with moderate or severe CRF present higher rates of major hemorrhage than those with mild to non-CRF during the next 12 days after VTE diagnosis, despite the administration of anticoagulant therapy [20,21].

Moreover, 1.5% of the patients had asthma, and 4.3% had chronic obstructive pulmonary disease (COPD). It has been shown that asthma increases the risk for PE. In comparison with the non-asthmatic people, asthmatic patients of all age groups run an increased risk for PE, which is even more increased depending on the age and the severity of the respiratory disease [22]. Even in a stable phase, COPD is considered an independent risk factor for PE. At the same time, a meta-analysis suggests that one out of four patients with a COPD exacerbation who need hospitalization may suffer from PE [18,22].

Diabetes mellitus (DM) appeared in 12.2% of the patients. Clinically, patients with PE who suffer from DM have a higher risk of mortality than those who do not suffer from DM, while it seems that elevated glucose rates increase the risk of VTE [23]. Also, a study on the Asian population considers insulin-independent diabetes as an independent risk factor for the development of DVT and PE [24].

15% of the hospitalized population were complicated with hemorrhage, of whom 18.4% experienced a major event. Major hemorrhages were fewer than minor (29.8% vs. 70.2%, *p* = 0.001). Hematuria was the most common hemorrhagic event. The overall prevalence of bleeding in acute PE cohorts is approximately 10/100 patient-years [25,26,27,28].

Specifically, in the MAPPET registry, among 1001 cases of PE, 92 (9.2%) presented a major hemorrhage required a transfusion of blood units or discontinuation of the anticoagulant therapy [29]. In the EMPEROR registry, 10.3% of the patients with massive pulmonary embolism (MPE) and 3.5% of those without MPE had hemorrhagic complications. 3 out of the 63 patients in the second group died because of the hemorrhage [7].

In the IPER registry, among 1716 patients with PE, a loss of hemoglobin > 4 g/dL was reported in 53 patients (3.1%), while 6 out of 10 patients with intracranial hemorrhage died [30]. In the ZATPOL registry, hemorrhagic complications were reported in 6% (67 out of 1112) of the patients with PE. Major hemorrhage was reported in 3.6% of the patients, while 0.5% had a fatal hemorrhage. Among the patients receiving anticoagulant therapy, 24% (29 patients) presented hemorrhagic complications. Specifically, 19% (23 cases) of the hemorrhages were major and 5% were (6 cases) minor. 38 hemorrhagic cases were reported in patients who had not received thrombolysis. 17 of them were major and 21 were minor. Among the 67 cases that presented hemorrhagic complications, 17 were presented after oral anticoagulant therapy was initiated [31].

Recently, a higher risk of bleeding (RR: 2.53, 95% CI: 1.60–4.00; I 2: 65%) has been reported in ICU patients receiving an anticoagulant therapeutic regimen [26,27]. In the elderly population, in which the risk of acute PE is increased due to advanced age, bleeding is even more pronounced, with the risk of major bleeding including intracerebral bleeding doubling in patients aged above 80 years and the risk of hemorrhagic complications is highest in the early days of treatment [32,33,34,35]. Interestingly, the risk of bleeding resulting in hospitalization or death within 3 and 12 months after the index PE admission increased over the last years [36].

Several bleeding risk prediction scores have been proposed, including the VTE-BLEED, RIETE, HASBLED, and HEMORR2HAGES scores [31,32,33,34,35], that have several limitations as most are retrospective, few focus on real-life cohorts, and patients in the stable (not acute) phase of anticoagulation are mainly included [37].

Most patients hospitalized due to PE received LMWH related to 6 major and 39 minor hemorrhagic episodes. Fondaparinux was only related to minor episodes of hemorrhage. According to studies that have compared it with enoxaparin, the percentage of major hemorrhage in 9 days is much lower when fondaparinux is used rather than enoxaparin [38]. On the other hand, we found that nadroparin related to 83.3% of the major events. Generally, it has been reported that absolute major bleeding rates are low for all LMWH agents [38]. Nevertheless, twice-daily dosing with nadroparin appeared to be associated with a 1.77 times greater bleeding risk as compared with once-daily dosing, as also suggested in a meta-analysis of controlled clinical trials [38,39].

As initial therapy, low-risk patients can receive DOACs, specifically rivaroxaban or apixaban [29]. Rivaroxaban and apixaban can be given in a higher initial dose without previous heparin therapy [29]. In the present study, 4.1% were receiving DOACs, and they underwent one major and one minor hemorrhagic episode. Multiple clinical studies support the safer bleeding profile of DOACs over Vitamin K antagonists [38]. However, it has been supported that DOACs at standard dose, except apixaban, had a higher risk of major gastrointestinal bleeding compared to warfarin. Apixaban had a lower rate of major gastrointestinal bleeding compared to dabigatran and rivaroxaban [40].

2% of patients with major events received a transfusion, and 11% were admitted to ICU. The bleeding events lasted for 3 ± 2 days. No death was recorded. Hemorrhagic complications were associated with an average hospitalization of 10.7 days, with higher risk of hospital-acquired infection and higher healthcare cost, compared to 7.4 days of hospitalization for those without bleeding, In-hospital major bleeding has been identified as strong predictor of in-hospital (OR 7.7, 95% CI 2.3–25.8) and 1-year mortality (HR 3.6, 95% CI 2.0–6.6), especially in normotensive patients [41]. Generally, an improvement in mortality has been reported over years attributed to both a real improvement in patient care and “over-diagnosis” of incidental and sub-segmental PE [36].

According to a recent meta-analysis of 14 randomized controlled trials and 13 cohort studies, including 9982 patients who received a vitamin K antagonist and 7220 received a DOAC, it has been supported that the incidence of major bleeding was statistically significantly higher among those who had creatinine clearance less than 50 mL/min [42]. Accordingly, in the present study, we found a correlation between high serum creatinine levels and hemorrhagic complications, but the regression model did not prove that this variable was an independent predictor of hemorrhage. A few limitations need to be noted regarding the present study. A major limitation of this study was its retrospective design that it might generate a great deal of missed data. There was also absence of data on potential confounding factors.

## 5. Conclusions

15% of the hospitalized patients of the study (49 patients) presented an episode of hemorrhage, while 18.4% of them presented an episode of major hemorrhage. Hemorrhages were mainly minor and there was no hemorrhage leading to death. 16.2% of the patients with hemorrhagic complication (2.1% of the total population) needed transfusion. The average duration of hemorrhage was 3 ± 2 days. 2.1% of the patients with major hemorrhage (0.3% of the total population) needed to be transferred to an ICU, because of the hemorrhagic complication. 83.3% of the cases that presented major hemorrhage and received LMWH were given nadroparin. There was not any independent predictor of hemorrhage, but there was a correlation between high Wells score or high levels of serum creatinine and hemorrhagic complication.

Only a few studies investigated the in-hospital hemorrhage cases in patients with PE, as detecting these rare events in large datasets remains difficult. The present study evaluating data throughout an 8-year period highlights a significant likelihood of bleeding and a small, but not negligible, possibility of major hemorrhage in patients hospitalized for PE. We found that nadroparin administration was associated with major hemorrhagic events; thus, it should probably not be the first therapeutic choice among other LMWH during the in-hospital treatment of patients with PE. Until now, there are no clear guidelines and scientific evidence available for physicians in this field for early diagnosis and tools to avoid hemorrhagic complications in patients hospitalized for PE. The optimal management of bleeding involves the application of predictive scores in combination with anticoagulant reversal strategies. However, risk assessment tools are relevant in managing patients with atrial fibrillation but are not widely validated in PE patients. Hence, the performance of existing prediction models in patients with PE should be further assessed. More comprehensively, the combination of clinical, biological, and genetic markers should be incorporated to build predictive scores to estimate the risk of bleeding and help the decision process about the proper type of anticoagulant treatment.

## Figures and Tables

**Table 1 jpm-12-01133-t001:** The parameters were extracted from the e-recordings of the patients hospitalized with PE.

Demographic Data	Medical History	Symptomatology, Clinical Picture, Estimation of Clinical Probability
Laboratory testing on admission and variation of laboratory parameters	The size of pulmonary emboli	Initial therapy
The burden of hemorrhagic episode	Anticoagulant therapy	Intensive care unit entrance, hospitalization length

**Table 2 jpm-12-01133-t002:** Demographics and comorbidities of the sample, *n* = 326.

Characteristics	Total n = 326	Males n = 188 (57.7)	Females n = 138 (42.3)
Age (years)	68.7 ± 17.0	64.9 ± 17.5	74.0 ± 15
Any comorbidity	281 (86.2)	162 (57.7)	119 (42.3)
No comorbidity	29 (8.9)	17 (58.7)	12 (41.3)
Malignancy	56 (17.1)	35 (62.5)	21 (37.5)
Lung cancer	12 (3.7)	12 (100)	0
First diagnosis of malignancy	9 (2.8)	6 (66.7)	3 (33.3)
History of thrombosis	82 (25.1)	52 (63.4)	30 (36.6)
Antiplatelet treatment	56 (17.1)	32 (57.1)	24 (42.9)
Previous hemorrhage	19 (5.8)	11 (57.9)	8 (42.1)
Thrombophilia	30 (9.2)	21 (70)	9 (30)

Note: Data are expressed as mean ± SD or as frequencies (percentages).

**Table 3 jpm-12-01133-t003:** Wells scores, Geneva scores, clinical and laboratory data on patients’ admission, *n* = 326.

Parameter	Total (*n* = 326)	Men (*n* = 188)	Women (*n* = 138)	*p*-Value
Wells score	5 ± 4	5 ± 4	5 ± 4	0.849
Geneva score	3 ± 4	3 ± 4	3 ± 4	0.955
PO2	70 ± 17	70 ± 19	69 ± 18	0.734
PCO2	34 ± 8	34 ± 9	34 ± 6	0.612
Platelets	257 ± 104	264 ± 111	248 ± 93	0.258
Urea	41 ± 20	42 ± 22	39 ± 16	0.161
Creatinine	1.17 ± 0.07	1.32 ± 2.3	0.9 ± 0.3	0.132
CRP	6.5 ± 6.4	6.3 ± 6.2	6.7 ± 6.1	0.641
D-dimer	2121 ± 1813	2083 ± 1722	2167 ± 1928	0.748
ΒΝP	4283 ± 4442	3995 ± 4414	4860 ± 5122	0.767
AST	37 ± 30	34 ± 21	40 ± 12	0.636
ALT	33 ± 20	32 ± 23	34 ± 22	0.814
HCT	39 ± 6	39.4 ± 5.4	39.2 ± 6.8	0.839

Note: Data are expressed as mean ± SD; Abbreviations: ALT, Alanine Aminotransferase; AST, Aspartate Aminotransferase; ΒΝP, Brain; Natriuretic Peptide; CRP, C-reactive protein; HCT, hematocrit; PO2, partial pressure of oxygen; PCO2, partial pressure of carbon dioxide.

**Table 4 jpm-12-01133-t004:** Anticoagulant therapy administered during patients’ hospitalization, *n* = 326.

Preparation	Frequency *n*, (%)
Low-Molecular-Weight Heparin (not specified)	132 (40.4)
Fondaparinux	73 (22.3)
Nadroparin	47 (14.4)
Enoxaparin	26 (8)
Tinzaparin	23 (7.1)
Classic heparin	1 (0.3)
Rivaroxaban	13 (4)
Apixaban	1 (0.3)
Dabigatran	2 (0.6)
Acenocoumarol	8 (2.4)
Total	326

Data are expressed as frequencies (percentages).

**Table 5 jpm-12-01133-t005:** The hemorrhagic sites of patients hospitalized due to PE, *n* = 326.

		Site of Hemorrhage						Total
		Gastric Bleeding	Hemoptysis	Bloody Sputum	Hematoma	Hematuria	Metrorrhagia	
Gender	Males	4 (18.1)	1 (4.5)	1 (4.5)	0	16 (72.7)	0	22
	Females	1 (3.7)	4 (14.8)	3 (11.1)	6 (22.2)	13 (48.1)	1 (3.7)	27
Total		5 (10.2)	5 (10.2)	4 (8.1)	6 (12.2)	29 (59.1)	1 (2)	49

Data are expressed as frequencies (percentages).

## Data Availability

The data that support the findings of this study are available on request from the corresponding author, O.S.K.

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
