# Peer review of "Prevalence of Hemorrhagic Complications in Hospitalized Patients with Pulmonary Embolism"

_jpm, 2022, doi:10.3390/jpm12071133_

Round 1

Reviewer 1 Report

 A very good , detailed presentation of the study conducted by the authors. It would be interest to add authors' opinion about any possible suggestions to further improve, in the future,  any aspect  (diagnosis, management, workflow, complications) of the parameters studied.

Author Response

A very good, detailed presentation of the study conducted by the authors. It would be interest to add authors' opinion about any possible suggestions to further improve, in the future,  any aspect  (diagnosis, management, workflow, complications) of the parameters studied.

Response to Reviewer 1 Comments:

Response 1: We sincerely thank you for your kind words about our paper. We are delighted to receive positive feedback from you. In the revision, we have provided some main conclusions and future directions regarding the management of PE patients (page 8, lines 311-326). We appreciate you taking the time to offer us your insights related to the paper.

Reviewer 2 Report

In the current research article, " Prevalence of hemorrhagic complications in hospitalized patients with pulmonary embolism ", the authors classified the occurrence of hemorrhagic complications in 326 hospitalized patients and showed a positive association between the use of anti-coagulants and the occurrence of hemorrhages. I have following comments/suggestions about the article:

Comments

Comment 1: Authors need to summarize the importance of their study. How their study will benefit the medical field. What are authors’ suggestions on improving the current treatment regimen or any pre-tests needed to avoid the problem of hemorrhages in PE patients?

Minor Edits:

1.     Please check spacing after paragraphs and make sure that it is consistent throughout the manuscript

2.     Line 294: 15% of hospitalized “patients” presented…

3.     Line 302: Why there is another conclusion section within the conclusions?

Author Response

Response to Reviewer 2 Comments:

In the current research article, " Prevalence of hemorrhagic complications in hospitalized patients with pulmonary embolism ", the authors classified the occurrence of hemorrhagic complications in 326 hospitalized patients and showed a positive association between the use of anti-coagulants and the occurrence of hemorrhages. I have following comments/suggestions about the article:

Comments

Comment 1: Authors need to summarize the importance of their study. How their study will benefit the medical field. What are authors’ suggestions on improving the current treatment regimen or any pre-tests needed to avoid the problem of hemorrhages in PE patients?

RESPONSE: Thank you for this great comment. In the revision, we summarized the importance of our study and have provided some main conclusions and future directions regarding the management of PE patients (page 8, lines 311-326). We appreciate you taking the time to offer us your insights related to the paper.

Minor Edits:

  1. Please check spacing after paragraphs and make sure that it is consistent throughout the manuscript

RESPONSE: Thank you for this point. We have checked the spacing, as suggested.

  1. Line 294: 15% of hospitalized “patients” presented…

RESPONSE: Thank you for this point. We have corrected it. We apologize for the omission.

  1. Line 302: Why there is another conclusion section within the conclusions?

RESPONSE: Thank you for the comment. We have deleted this repetition.

We appreciate you taking the time to offer us your insights related to the paper. We found your feedback very constructive. We tried to be responsive to your concerns.